Quality assessment of a serum and xenofree medium for the expansion of human GMP-grade mesenchymal stromal cells

Aussel Clotilde 1
Busson Elodie 2
Vantomme Helene 2
Peltzer Juliette 1
Martinaud Christophe christophe.martinaud@inserm.fr 2
1 Biomedical Research Institute of the Armed Forces , Clamart , France
2 Advanced Therapy Medicine Unit, French Military Blood Institute , Clamart , France
Mezey Eva
Electronic publication date: 2022 May 30
Publication date: 2022
Volume: 10
Electronic Location ID: e13391
Received 2021 Jan 5; Accepted 2022 Apr 15
Copyright: ©2022 Aussel et al.
Copyright year: 2022
Copyright holder: Aussel et al.
License: This is an open access article distributed under the terms of the Creative Commons Attribution License, which permits unrestricted use, distribution, reproduction and adaptation in any medium and for any purpose provided that it is properly attributed. For attribution, the original author(s), title, publication source (PeerJ) and either DOI or URL of the article must be cited.
License URL: https://creativecommons.org/licenses/by/4.0/

Keywords: Mesenchymal stromal cells, Platelet Lysate, GMP medium, Clinical grade production, Quality

Funding: The authors received no funding for this work.

==============================
Background

Cell-based therapies are emerging as a viable modality to treat challenging diseases, resulting in an increasing demand for their large-scale, high-quality production. Production facilities face the issue of batch-to-batch consistency while producing a safe and efficient cell-based product. Controlling culture conditions and particularly media composition is a key factor of success in this challenge. Serum and Xeno-Free Media (SXFM) represent an interesting option to achieve this goal. By reducing batch to batch variability, they increase Good Manufacturing Practices (GMP)-compliance and safety regarding xenogenic transmission, as compared to fetal bovine serum (FBS) supplemented-media or human platelet lysate supplemented medium.

Methods

In this study, the isolation, expansion and characteristics including the anti-inflammatory function of human mesenchymal stromal cells (MSC) are compared after culture in MEMα supplemented with human Concentrate Platelet Lysate (hCPL, reference medium) or in MSC-Brew GMP Medium. The latter is a GMP SXFM manufactured in bags under strictly controlled conditions in volumes suitable for expansion to a clinical scale and does not require neither pre-coating of the cell culture units nor the addition of blood derivatives at the isolation step.

Results

We showed that MSC derived from human bone-marrow and adipose tissue can be successfully isolated and expanded in this SXFM. Number and size of Colony-Forming Unit fibroblast (CFU-F) is increased compared to cells cultivated in hCPL medium. All cells retained a CD90+, CD73+, CD105+, HLADR−, CD34−, CD45− phenotype. Furthermore, the osteogenic and adipocyte potentials as well as the anti-inflammatory activity were comparable between culture conditions. All cells reached the release criteria established in our production facility to treat inflammatory pathologies.

Conclusions

The use of MSC-Brew GMP Medium can therefore be considered for clinical bioprocesses as a safe and efficient substitute for hCPL media.

Introduction

Human mesenchymal stromal cells (MSC) are multipotent adult stem cells and progenitors isolated from tissues of mesodermal origin. Bone marrow (Friedenstein et al., 1974) and adipose tissue (Zuk et al., 2001) sources are widely used for MSC clinical grade production. MSC are defined by the International Society for Cell & Gene Therapy consortium as plastic adherent cells, expressing positive and negative surface markers, and multilineage differentiation potential towards osteoblast, adipocyte and chondrocyte lineages (Dominici et al., 2006). Many studies have highlighted their immunomodulatory, anti-apoptotic, anti-oxidative, pro-angiogenic, regenerative or anti-fibrotic properties mediated by direct cellular contact or by paracrine activity (Spees, Lee & Gregory, 2016). These properties make MSC a very interesting tool for regenerative medicine. They are currently the most evaluated cells in clinical trials worldwide, specifically in four main areas: neurological, joint, cardiovascular and graft versus host diseases (Kabat et al., 2020). Moreover, a few of them have already received a marketing authorization in several countries.

Oversight agencies have driven legal frameworks and definitions for expanded MSC used for clinical applications. Tissue engineered medicinal products (TEMPs) (Regulatory Considerations for Human Cells, Tissues, and Cellular and Tissue-Based Products: Minimal Manipulation and Homologous Use, 2020) and advanced therapy medicinal products (ATMPs) (Smith, 2015) have been introduced more than a decade ago in the USA and in Europe respectively. Since then, robust and standardized processes as well as specific quality controls have been set up to ensure the efficacy and safety of MSC therapies. However, there are some limitations that need to be addressed to improve batch-to-batch consistency and prevent suboptimal clinical outcomes due to lack of complete knowledge of cell state (Galipeau, 2013). Controlling culture conditions is currently one of the main challenges for production facilities (Brown et al., 2019).

It is widely accepted that different culture conditions may affect MSC properties (Liu et al., 2017). In particular, modifications of the transcriptome, proteome, secretome or cellular organization could affect MSC engraftment and efficiency upon transplantation (Tonti & Mannello, 2008). These cells are extremely sensitive to their environment and require, during ex-vivo cell processing, that all production parameters are clearly defined (media composition, temperature, oxygenation, seeding density, duration of expansion, two-or three-dimensional cultures, and so on). Media composition is one of the trickiest parameters to manage in order to control culture conditions. MSC are historically cultured in medium supplemented with fetal bovine serum (FBS) which carries the risk of transmitting prions or viruses and triggering immune responses (Spees et al., 2004). To reduce this risk, human concentrate platelet lysate (hCPL) has been used as a suitable alternative to FBS in numerous clinical grade ATMPs (Doucet et al., 2005; Bieback et al., 2009; Aldahmash et al., 2011).

Therefore, hCPL is a source of non-xenogenic growth factors that prevent xeno-immunization, xenopathogen transmission and ethical considerations regarding animal welfare (Schallmoser et al., 2007; Burnouf et al., 2016). Our institute is authorized by French regulatory agencies to produce its own hCPL from platelet apheresis concentrate.

However, a quarantine period (minimum of 2 months) before use is required for each produced batch. Since February 2018, in France, all platelet concentrates undergo a pathogen attenuation process based on the use of a psoralen derivatives (amotosalen/Intercept®; Cerus, Concord, CA, US) to reduce the risk of transmission of potential microbial contaminants. This treatment eliminates the need to quarantine these biological products. This method was approved by the French National Drug Agency in July 2005, but some studies revealed that this inactivation process result in minor functional alterations of platelets such as a decrease in their aggregation or a death by apoptosis of these inactivated platelets (Stivala et al., 2017). In addition, psoralen amotosalen, even after filtration, could be present at a residual level and to our knowledge no data are available to address the impact of this process on the properties of expanded MSC.

Another way to eliminate potential pathogens is to gamma irradiate hCPL before use (Viau et al., 2019). However, no pathogen inactivation process has been shown to eliminate infectivity from all pathogens.

In addition to contamination risks, the issue of batch to batch variability for both FBS and hCPL leads to high variations in MSC expansion and functionality (Bieback et al., 2019).

Each batch consequently needs to be functionally tested at reception as every raw material used in the MSC process. To overcome these shortcomings, different strategies can be implemented such as preparing hCPL from a batch of more than 200 donors to improve the homogeneity of MSC productions compared to batches of few donors (Viau et al., 2019). These kinds of batches are mostly prepared from expired platelet concentrates initially intended for human transfusion, gamma-irradiated to reduce the risk of transmission of pathogens due to the high number of donors, as recommended by the European Pharmacopia (Chapter 5.2.12) and filtered resulting in a potential loss of expansion capacity of MSC cultivated with these platelet lysates when compared to non-treated and non-filtered hCPL.

At last, hCPL quality controls and release criteria that have been recently proposed for hCPL producers (such as osmolality, pH, total protein, platelet-derived growth factors dosage, etc.) may be challenging for small structure and encouraging alternative products (Schallmoser et al., 2020).

Recently, many companies have developed serum and xenogenic free media (SXFM) enabling optimized and reproducible MSC production with high quality components and testing of each batch for its functional performance. As highly encouraged by the regulators to have a backup media in all Good Manufacturing Practices (GMP) processes (Guideline on human cell-based medicinal products, EMEA/CHMP/410869/2006), the aim of this study was to validate the use of a GMP standardized SXFM medium: the MSC-Brew GMP Medium (Miltenyi Biotec®, Bergisch Gladbach, Germany). To our knowledge, this is the only SXFM in the market that is manufactured under GMP compliance, conditioned in bags to allow culture in closed system (as authorized in our unit), that does not require any coating of the culture surfaces and claims to allow isolation and expansion of MSC from bone-marrow, adipose tissue and umbilical cord tissues.

We used human-derived platelet lysate-based supplemented MEM α as a reference. We performed all the required quality controls in accordance with the European legislation concerning advanced therapy medicinal products for intermediate product and batch release.

Materials & Methods

MSC isolation, culture and proliferation study

Human Bone-Marrow-derived MSC (BM-MSC) were obtained from consenting donors (n = 4, Table 1) undergoing routine total hip replacement surgery or bone-marrow aspiration from iliac crest in Percy Military Medical Center (Clamart, France). Mononuclear cells (MNCs) were plated at a density of 200,000 cells/cm2 (according to our clinical grade MSC production protocol) in 25 cm2 flasks and cultured at 37 °C in 95% air and 5% CO2.

Table 1 MSC donor’s demographic data.

	MSC source	
	Bone-marrow	Adipose tissue	
Donor	Sex	Age (Years)	Sex	Age (Years)	
1	Female	51	Female	40	
2	Male	19	Female	49	
3	Female	47	Male	42	
4	Male	27	Female	22	

Adipose tissue-derived MSC (Ad-MSC) were collected from consenting donors undergoing a liposuction (n = 4, Percy Military Medical Center). Adipose tissue was washed with PBS (Macopharma, Tourcoing, France) supplemented with 10 µg/ml Ciprofloxacine (Panpharma, Boulogne-Billancourt, France) then digested in 0.3U/ml GMP-Collagenase NB6 (Nordmark, Germany) for 45 min at 37 °C. The enzymatic reaction was then stopped using Minimum Essential Media or MEMα (Macopharma, Tourcoing, France) supplemented with 10% human albumin (LFB, Les Ulis, France); After centrifugation, the stroma vascular fraction was washed with medium, and isolated mononuclear cells were seeded at 10,000 cells/cm2 in 25 cm2 flasks and cultured at 37 °C in 95% air and 5% CO2.

For each source of MSC, two types of culture media were used and compared to isolate and expand MSC: hCPL and SXFM. (1) hCPL medium consisted of clinical grade MEMα (Macopharma, Tourcoing, France), 2UI/mL Sodium Heparin (Sanofi, Paris, France), 8% human clinical platelet lysate. Clinical hCPLs were obtained from the Military Blood Institute (Clamart, France) as previously described (Doucet et al., 2005). Briefly, platelet concentrates from apheresis of single donors were aliquoted within the 5 days following donation, frozen at −40 °C and thawed extemporaneously at 37 °C for 25 min before addition to the medium. Each batch (n = 5) used was qualified according to release criteria, including serologic tests on donor 2 months after the donation as well as sterility, mycoplasma and a functional assay based on the capacity of the batch to expand MSC as compared to bovine fetal serum. Neither centrifugation nor filtration was performed on them. (2) SXFM (MSC-brew GMP medium Miltenyi Biotec®, Bergisch Gladbach, Germany) was composed of MSC-Brew basal medium and MSC-Brew GMP Supplements I and II that were frozen and added extemporaneously. No supplemental information on its composition was available except those provided by the specification sheet mentioning salts, amino acids, vitamins, buffers, human proteins and TFG-B1 as main components. No heparin was added to this medium. Three different batches were used.

After 1 or 2 days, the non-adherent cells from the two different sources were removed and the cultures were fed with fresh medium. Thereafter, medium was changed once a week. When MSC reached 80–90% of confluence they were trypsinised with a recombinant trypsin (Trypzean®, Sigma-Aldrich, St. Louis, MO, USA). Viable cells were enumerated with Trypan Blue staining and seeded at 4,000 cells/cm2 in a new culture flask. Passages (P) were done after 3 to 6 culture days depending on the MSC sources and patients. The culture was stopped at the end of the passage 2 because no cells beyond this passage are injected into our clinical protocols.

Cumulative cell expansion number was based on the number of cells obtained at the end of each passage multiplied by the expansion rate. The expansion rate was obtained by dividing the total number of confluent cells by the number of seeded cells.

The flowchart of the MSC production and quality controls performed at each culture step is summarized in Fig. 1.

Figure 1 MSCs’ isolation and production flow chart.

Diagram of production and controls carried out on the process and final products.

Colony forming unit-fibroblastic (CFU-F) assay

CFU-F formation was examined on 200 MSC harvested at the end of passages 0, 1 and 2. Cells were cultured in 25 cm2 flasks with either hCPL or SXFM. Once a week, the medium was replaced and cultures were stopped on day 10. Then, the cell layer was fixed and stained with a Crystal violet dye for 20 min (Sigma Aldrich, St. Louis, MO, USA). Clonogenic efficiency was calculated as the number of colonies bigger than 50 cells in a binocular magnifier (x2) by two different technicians.

Phenotypic assay

For MSC phenotypic characterization, a multi-parameter analysis was carried out by flow cytometry (Navios, Beckman Coulter®, Brea, CA, USA) on cells harvested at the end of passage 1 and 2. After trypsin detachment, cells were centrifuged and incubated in a buffer containing PBS, 2% human serum albumin (LFB, Les Ulis, France) and 2 µg/mL polyvalent human immunoglobulin (R&D Systems, Minneapolis, MN, USA) to minimize non-specific antibody binding. Cells were then incubated with fluorescent monoclonal antibodies at a saturating concentration (5 µL per 105 cells): CD90, CD73, CD105, CD45, HLA-DR, CD146, CD34 and isotypic controls (Iotest, Beckman Coulter, Brea, California, USA) for 20 min at 4 °C. After 2 PBS washes, analyzes were carried out by flow cytometry (Navios, Beckman Coulter®). The FlowJo software was used to analyze 10,000 stained cells. The percentage of positive stained cells was calculated in comparison to the isotypic control after gating on FS/SS parameters.

Multipotency assay

MSC potential to differentiate into an osteogenic and adipogenic lineage was evaluated at the end of the passage 1. MSC previous amplified in hCPL or SXFM were seeded at 3,000 cells/cm2 in MEMα (Biologicals Industries, Cromwell, CT, USA) supplemented with different components for osteogenic stimulation: 100 nM dexamethasone (Sigma Aldrich, St. Louis, MO, USA), 10 mM β-glycerophosphate (Sigma Aldrich, St. Louis, MO, USA), 0.05 mM ascorbic acid (Sigma Aldrich, St. Louis, MO, USA), 10 µg/ml ciprofloxacin (Panpharma, Boulogne-Billancourt, France) and 10% fetal bovine serum (Hyclone, San Angelo, TX, USA). Unstimulated MSC were used as controls. After 21 days of incubation at 37 °C in a humidified atmosphere (5% CO2), the cells were fixed with ethanol and then various staining were carried out.

First, alkaline phosphatase activity was evaluated using a colorimetric assay (Vector Blue Alkaline Phosphatase Substrate Kit III, Vector laboratories, Burlingame, California, USA). Alizarin red (Merck, Darmstadt, Germany) and Von Kossa staining (Sigma Aldrich, St. Louis, MO, USA) were also used to identify the mineralized bone matrix in cells.

Adipogenic differentiation was induced in MSC subconfluent cultures by three treatment cycles in DMEM with a high glucose concentration (4.5 g/L) (Gibco, Billings, MT, USA) supplemented with 10 mg/ml insulin (Sigma Aldrich, St. Louis, MO, USA), 0.5 mg/ml dexamethasone (Sigma Aldrich, St. Louis, MO, USA), 28 mM indomethacin (Sigma Aldrich, St. Louis, MO, USA), 450 mM IBMX (Sigma Aldrich, St. Louis, MO, USA), 10 µg/ml Ciprofloxacine (Panpharma, Boulogne-Billancourt, France) and 10% fetal bovine serum (Hyclone, San Angelo, TX, USA). Cycles were performed by induction during 3 days followed by 1 day of maintenance of the cultures in DMEM high glucose (Gibco, Billings, MT, USA), 10% fetal bovine serum (Hyclone, San Angelo, Texas, USA) and 10 mg/ml insulin (Sigma Aldrich, St. Louis, MO, USA) alone. After 3 weeks of incubation, cultures were histochemically stained with oil red O allowing lipid droplets detection (Cayman Chemical, Ann Arbor, MI, USA). Microscopic pictures were taken using a Leica invert microscope.

Immunomodulary assay

To assess immunomodulatory potency of MSC, they were co-cultured, after trypsinization at the end of passage 1, with stimulated promonocytes (THP-1 cell line). First, THP-1 (ATCC® Manassas, Virginie, USA) cells were maintained between 0.2 and 1 × 106 cells/mL in RPMI 1640 Glutamax™ (Gibco, Billings, MT, USA), 10% FBS (Hyclone, San Angelo, Texas, USA), 1% penicillin/streptomycin (Gibco, Billings, MT, USA) and 0.05mM β-mercaptoethanol (Gibco, Billings, MT, USA). The cells were fed every three to four days with new medium to reach their conventional seeding density. Then, production of pro-inflammatory cytokine TNF-α or anti-inflammatory cytokine IL-1ra was evaluated after stimulation with lipopolysaccharide (LPS) of THP1 co-cultured or not with MSC as described by Zhang et al. (2010). Briefly, THP-1 cells were seeded at 170,000 cells/mL in one mL of complete medium in a 6-wells plate, MSC were added at 1:1 ratio, the coculture was exposed to 1 µg/mL of LPS for 24 h (Sigma Aldrich, St. Louis, MO, USA). Unstimulated controls were carried out with or without MSC. Supernatants of each condition were then collected and centrifuged. TNF-α and IL-1ra levels were evaluated by immunoassay (DuoSet® ELISA Kits, Bio-Techne, Minneapolis, Minnesota, USA).

Statistical analyses

All data are expressed as median with interquartile range (IQR). Statistical comparisons for proliferation, phenotype and clonogenicity are calculated using the non-parametric Wilcoxon test for the comparison of the two groups (hCPL versus SXFM). Immunomodulary assays were analyzed with the non-parametric Friedman test to compare the 3 groups: THP-1 + LPS vs THP-1 + LPS + MSC in hCPL vs THP-1 + LPS + MSC in SXFM.

Statistical analyses were performed on Prism 7 Graphpad software. A p-value < 0.05 was considered significant. A minimum of three independent experiments were performed each time.

Protocol approval and patient consent

This study was approved by the Scientific and Ethic Board of the French Military Blood Institute (IRB 0001-2017). All volunteers who were asked to participate in the study consented. Samples from patients were collected from operatory waste in the operating room and were fully anonymized before experimentation. Patients were informed according to French regulation through medical book and hospital policy, and written consent was recorded.

Results

Morphology, proliferation and cell viability

BM-MSC and Ad-MSC were successfully isolated in SXFM. They had a typical MSC fibroblastic morphology under phase-contrast microscopy (Fig. 2A). In MSC cultured in hCPL, we could also observe round and refractive cells/deposits, due to the accumulation of platelet lysate debris and the presence of macrophages (round cells with the characteristic “fried egg” shape morphology staying adherent after trypsin action). Otherwise, MSC morphology was quite similar in the two media, even though MSC cultured in hCPL tended to be more fusoidal.

Figure 2 Morphology and proliferation.

(A) BM-MSC and Ad-MSC morphology assessment (passage 2) using a phase-contrast microscope (scale bar = 200 µm) (B) Cumulated cell expansion number over 3 passages. Data are presented as median ± interquartile range (n = 4, ns = non-significant, Wilcoxon test was used to compare medians at P2).

To assess the impact of the media on cell division, we calculated the cumulative cell expansion number at each passage. We did not observe any significant difference between SXFM or hCPL up to P2 (Fig. 2B). The total number of BM-MSC at P2 reached 8.09 × 106 [IQR 71.1 × 106] cells in hCPL vs 7.21 × 106 (IQR 47 × 106) in SXFM (n = 4, p = 0.25). The total number of Ad-MSC at P2 reached 6.49 × 107 (IQR 1.22 × 109) cells in hCPL vs 6.19 × 107 (IQR 8.02 × 108) in SXFM (n = 4, p = 0.625). Cell viability was over 90% in both media, at each passage (File S1). Therefore, SXFM allowed an effective expansion of MSC, similar to that achieved with classic hCPL medium without allowing the adherence of contaminating cells such as macrophages.

CFU-F efficiency

We then compared the CFU-F efficiency of cells cultured in hCPL or SXFM. The macroscopic observation of cells stained with crystal violet allowed us to notice that the size and the density of colonies seemed increased in the SXFM condition compared to the hCPL condition with the two MSC sources (Fig. 3A). The analyses showed that the number of CFU-F obtained on three passages was significantly higher when cells were cultured in SXFM compared to hCPL regardless of the source of MSC (Fig. 3B). Indeed, for BM-MSC, the median of the number of colonies was 23.63 [IQR 20.87] vs 40.25 [IQR 36.88] in hCPL vs SXFM respectively (n = 12, p = 0.0137). For Ad-MSC, the median of the number of colonies was 18.63 [IQR 41.25] vs 67.38 [IQR 43.44] in hCPL vs SXFM respectively (n = 12, p = 0.0049).

Figure 3 CFU-F efficacy.

(A) Aspect of the flasks of the clonogenicity test and individual colonies of BM-MSC or Ad-MSC at P1 captured with a binocular magnifier (scale bar = 1 mm) (B) Colonies containing 50 or more MSC were scored among 3 passages as CFU-F colonies. Data are presented as median ± interquartile range (n = 12, ∗p < 0.05 and ∗∗p < 0.01, Wilcoxon test was used to compare medians obtained for values of the 3 passages).

Phenotypic characterization

At the end of P1 and P2, cells were screened for MSC typical surface markers by flow cytometry (Fig. 4B). MSC from both sources and media exhibited a high expression (≥90% positive cells) of CD90, CD73, CD105 and a negative or very low expression of CD45, CD34 and HLA-DR (≤ 5% positive cells). Interestingly, SXFM increased significantly the CD146 expression level compared to hCPL: 90.54 [IQR 8.33] in SXFM vs 31.80 [IQR 46.18] in hCPL (n = 8, p = 0.0156) (Fig. 4A).

Figure 4 Phenotype characterization of MSC.

(A) Data are presented as the median ± interquartile range of the percentage of positive cells for each marker for P1 and P2. (B) Radar diagram showing the median percentage of cells positive for different markers (n = 8 for Ad-MSC and n = 6 for BM-MSC, ∗p < 0.05, Wilcoxon test was used to compare medians obtained for values of the 2 passages).

Multipotency

The influence of SXFM on MSC multipotency was investigated after appropriate inductive stimulation at the end of passage 1 (Fig. 5). As shown in Fig. 5, cells cultured in hCPL and SXFM were strongly positive for mineralization markers alizarin red (Fig. 5A) and Von Kossa (Fig. 5B) and also for alkaline phosphatase activity (Fig. 5C) (strongly expressed by osteoblasts) while control cells were not. The adipocytic differentiation was demonstrated by oil red O staining of cytoplasmic lipid droplets (Fig. 5D). Both BM-MSC and Ad-MSC expressed lipid droplets. Altogether, cells cultured in SXFM retained their multipotency compared to cells cultured in hCPL, as shown by the osteogenic and adipogenic differentiation potential of MSC.

Figure 5 Multipotency.

Osteogenic differentiation potential of BM-MSC and Ad-MSC at P1 was examined with (A) Alizarin red or (B) Von Kossa and (C) alkaline phosphase activity stain. (D) Microscopic observation of BM-MSC induced or not (control) in the adipogenic direction and stained with oil red O.

Immunomodulary potency

We then evaluated the impact of the media on the immunomodulary potency of MSC. We co-cultured MSC harvested at the end of P1 and P2 with THP-1 pro-monocytes and stimulated the secretion of TNF-α with LPS. We used monocultures of THP-1 cells as a control. With BM-MSC in hCPL media, the secretion of TNF-α was significantly decreased in the co-cultured condition compared to the control well with THP-1 alone stimulated with LPS: 132.4 [IQR 98.72] vs. 452.3 pg/ml [IQR 448.5] respectively (n = 7, adjusted p value = 0.0099) and also trend to decrease with BM-MSC in SXFM: 268.9 [IQR 112.6] (Fig. 6A).

Figure 6 Immunomodulary assay for MSC potency.

Supernatant of THP-1 cells co-cultured with MSC at P1 and P2 was assayed for TNF-α and IL-1ra secretion. The cytokine concentration is normalized by the one measured in the LPS-treated THP-1 monoculture . Data are presented as median ± interquartile range with n = 7 for TNF- α secretion by BM-MSC, n = 6 for IL-1ra secretion by BM-MSC, n = 5 for TNF- α secretion by Ad-MSC and n = 4 for IL-1ra secretion by Ad-MSC. Friedman test was used to compare the effect of MSC expanded in the 2 media (hCPL and SXFM) to THP-1 stimulated by LPS, ns = non-significant, ∗p < 0.05 and ∗∗p < 0.01.

The secretion of IL-1ra was significantly increased in the presence of BM-MSC previously expanded in SXFM: 1263 [IQR 1713.6] vs. 270.9 [IQR 643.5] respectively (n = 6, adjusted p value = 0.0117) (Fig. 6B) and also trend to increase with BM-MSC in hCPL: 664.8 [IQR 849]. In addition, similar but non-significant trends were observed with MSC isolated from adipose tissue (Figs. 6C and 6D).

Discussion

The media currently used in clinic, as an alternative to media containing FBS, are xeno-free media including hCPL or industrial human plasma derivatives. hCPL presents batch to batch variations that can be reduced by production of large pools of donors per batch but that consequently increase the risk of infection by known or emerging human viruses. Viral inactivated GMP hCPL are nowadays largely distributed commercially, including among others: Human Platelet Lysate (Life Science Production), Platelet Lysate MultiPLi (Macopharma), PLUS™ GMP grade (Compass Biomedical, Inc). But it raises the question of platelet donations supplies that need to be sufficient to cover the needs for transfusion and advanced therapy medicinal products for which clinical trials are rapidly growing, as in France expired platelet concentrates are not suitable for clinical purposes.

In addition to hCPL, many commercially FBS free medium are available but, even if some are “GMP-compatible”, most of them are for research use only and not intended for use in therapeutic procedures. Among these research grade products, chemically defined media composed of only known products has been developed such as BD Mosaic™ Mesenchymal Stem Cell Serum-Free media (BD Biosciences) or Xerum free defined cell culture (TNC Bio BV). However, the main issue with chemically defined medium is the lack of cell growth (Gottipamula et al., 2016; Matthyssen et al., 2017) and attachment factors. Indeed, the use of chemically defined media implies precoating the plastic surfaces to support cell attachment and growth (Jung et al., 2010). Moreover, some media supplemented with human plasma derivatives are also not able to sustain MSC isolation without coating, for instance StemPro MSC SFM™ (Life Technologies) or MesenCult-ACF plus medium (Stemcell Technologies) (Hartmann et al., 2010; Patrikoski et al., 2013; Gottipamula et al., 2016). Regarding GMP products, some are not xeno-free as it is the case for TheraPEAK™ MSCGM™ (Lonza) or UltraCulture (Lonza) that contains bovine proteins including transferrin and albumin. They also generally require pre-coating such as the PRIME-XV™ MSC Expansion SFM or XSFM (Fujifilm) or StemPro_MSC (Life Technologies) in which cells show low adhesion and need the presence of fibronectin (Patrikoski et al., 2013; Heathman et al., 2015; Cimino et al., 2017) and that do not allow the isolation step of MSC without the addition of human serum AB or FBS in the first passage (Hartmann et al., 2010; Chase et al., 2012; Wuchter et al., 2016).

Because coating increases production complexity, time and cost, we chose to investigate a serum, xeno and coating-free media: MSC-Brew GMP medium (Miltenyi Biotec, Germany). To our knowledge this study is the first to evaluate MSC expansion in this media. This new type of medium is promising to allow MSC isolation without serum addition, control variability of GMP-MSC expansion (batch to batch reproducibility), as well as increase GMP compliance and safety regarding xenogenic transmission. It has the advantage of being packaged in bags, which is an important choice criterion for ATMP units because it is compatible with closed system culture procedures and, also, of having a research grade version (StemMACs MSC Expansion Media kit, XF; Miltenyi Biotec) with the same formulation, simplifying translational research.

Here, we showed that similarly to our reference medium (hCPL), SXFM sustained isolation, proliferation and maintained the main properties of MSC from two commonly used sources, bone marrow and adipose tissue. We decided to assess the effect of SXFM on proliferation rate, phenotype, CFU-F and, as a potency assay, the immunomodulation of a macrophages cell line. The studied parameters were evaluated on a small number of passages, without the possibility of attesting to the absence of difference which could occur at later passages. However, we chose to evaluate the parameters up to passages conventionally authorized in clinical trials. Proliferation rate is a key issue in cell-based therapy for regenerative medicine as up to 4 × 106 cells per kg are commonly required. Both media tested stimulated MSC growth up to P2. There was no significant difference in proliferative potential between the two culture conditions. In both conditions, the number of cumulative cell expansion obtained was sufficient to reach the necessary therapeutic dose. We notice that even if the cells isolated from the two tissue sources were not seeded at the same density at P0, the number of Ad-MSC obtained was more abundant in adipose tissue due to a quicker proliferation than the BM-MSC. This disparity in expansion rate is a difference usually found in both our experience and the literature (Berebichez-Fridman & Montero-Olvera, 2018). We evaluated the phenotype of MSC according to the position statement of the International Society for Cell & Gene (ISCT) regarding the minimal criteria for defining MSC (Dominici et al., 2006), stating that a minimum set of markers is required to characterize MSC. In clinic, specifications can differ within passages, sources and protocols: positive markers are usually considered valid when more than 90 to 95% of cells are positive for them, while negative markers are validated when they were express by less than 10 to 2% of cells. Both BM-MSC and Ad-MSC successfully expanded in SXFM and hCPL reached acceptance criteria at P1 for positive markers CD90, CD105 and CD73 as well as for the negative ones CD45 and HLA-DR. Unexpectedly, this study seemed to highlight an increase of CD146 positive cell number in SXFM cultivated Ad-MSC cells compared to cells cultured in hCPL. Interestingly, CD146+ MSC have been identified to have a higher capacity for supporting hematopoietic progenitor cells than unsorted bone marrow mononuclear cells in vitro (Sorrentino et al., 2008) and they exhibit the ability to reorganize the hematopoietic microenvironment into heterotopic sites in vivo (Sacchetti et al., 2007). In addition, it was demonstrated that CD146+ MSC’s subpopulation held greater migration potential towards a degenerated intervertebral disc model (Wangler et al., 2019). Finally, a positive correlation was already demonstrated between MSC clonal capacity, trilineage potency and CD146 expression (Sorrentino et al., 2008; Russell et al., 2010).

This is in line with our observations showing that SXFM trend to improved clonogenic efficiency (number of colonies) of Ad-MSC compared to hCPL medium. However, the ability of MSC to differentiate into osteoblasts and adipocytes was not different between the two media. A quantitative analysis of the multilineage differentiation potential of MSC could perhaps reveal more subtle differences between the conditions. Finally, we evaluated the impact of SXFM in a potency assay. For several years, the international scientific community has highlighted the difficulty of characterizing MSC, and therefore recommended to demonstrate their functionality, in order to assess their biological activity. Hence, potency assays are now mandatory for ATMP quality control. The first clinical grade MSC production authorization obtained by our cell therapy unit concerned radiological burns. Severe radiological burns are essentially characterized by the development of skin, muscle and bone tissue necrosis, progressing in successive inflammatory waves difficult to predict. We have shown a beneficial effect of MSC administration on musculo-cutaneous engraftment in irradiated patients, in particular through the reduction of systemic inflammation. Therefore, the functional test of inflammation in vitro, serves as a benchmark test and has been validated in the regulatory file provided to the health authorities. Our results highlight the potential of MSC to reduce LPS-induced TNF-α release by monocytes (THP-1 cell line) and to increase IL-1ra secretion whatever the culture medium used.

Conclusions

Altogether, for diseases requiring immunomodulatory property of MSC, MSC-Brew GMP media appears to be a convenient alternative to hCPL. Its use can be considered for clinical bioprocesses but will require further validations in the case of non-inflammatory clinical applications.

Supplemental Information

Supplemental Information 1 Raw data

Click here for additional data file.

Supplemental Information 2 Viability of Mesenchymal Stromal Cells during passages depending on supplementation

Viability is expressed as a percentage of viable cells among total cells, from passage 0 (P0) to passage 2 (P2). BM-MSC: bone marrow Mesenchymal Stromal Cells; Ad-MSC: adipose derived Mesenchymal Stromal Cells; hCPL: human Concentrate Platelet Lysate; SXFM: serum and Xeno-free Media.

Click here for additional data file.

We warmly thank surgeons who provided us raw material as well as the Miltenyi company that provided us the media. We are also grateful to Dr. Alexandra Seguin (University of Utah) who proceed to the proof-reading of the article.

Additional Information and Declarations

Competing Interests

Author Contributions

Human Ethics

Data Availability

The authors declare there are no competing interests.

Clotilde Aussel performed the experiments, analyzed the data, prepared figures and/or tables, authored or reviewed drafts of the paper, and approved the final draft.

Elodie Busson conceived and designed the experiments, analyzed the data, prepared figures and/or tables, authored or reviewed drafts of the paper, and approved the final draft.

Helene Vantomme performed the experiments, analyzed the data, prepared figures and/or tables, and approved the final draft.

Juliette Peltzer conceived and designed the experiments, analyzed the data, authored or reviewed drafts of the paper, and approved the final draft.

Martinaud Christophe conceived and designed the experiments, authored or reviewed drafts of the paper, and approved the final draft.

The following information was supplied relating to ethical approvals (i.e., approving body and any reference numbers):

The Scientific and Ethic Board of the French Military Blood Institute gave its approval to this study (IRB 0001-2017).

The following information was supplied regarding data availability:

The raw data are available in the Supplementary File.

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
