# Peer review of "Quality assessment of a serum and xenofree medium for the expansion of human GMP-grade mesenchymal stromal cells"

_PeerJ, doi:10.7717/peerj.13391_

## Round 0.1 · original submission · Major Revisions

Please, respond to the questions, comments of the reviewers. I strongly agree with Reviewer 1's comment that measuring TNFalpha alone does not clearly indicate the change of the monocytes/macrophages towards an anti-inflammatory character. IL10 and/or PGE2 measurement would be necessary to show the switch. Furthermore, the number of samples/donors in the individual Figures need to be clarified. N can not be 2 - that is not valid sample number for calculating significance.

Reviewer 1 ·

Basic reporting

The background of the figures (Fig5.d) demonstrating adipogenic differentiation is too dark and blue. It would be better to increase the intensity of illumination to get images with lighter background.

Experimental design

The work described in the manuscript of Martinaude et al. entitled “Quality assessment of a serum and xenofree medium for the expansion of human GMP-grade mesenchymal stromal cells” (#56142) is not novel. The authors perform experiments that have been recommended by the ISCT and published earlier. The manufacturer of this GMP medium (Milteny Biotech: MSC-Brew GMP Medium) have tested this product as described on the homepage. I agree that validation of a newly introduced serum and xenofree culture medium/bag in their stem cell production facility is absolutely necessary but this is of limited scientific value.
For demonstration of the immunosuppressive property of these clinical grade MSCs in co-culture with THP-1 cells, the authors measure the decrease in the amount of the pro-inflammatory cytokine TNFalpha. This experiment alone is not the most adequate choice, especially if the authors show only ratio and no absolute numbers. The TNFalpha absolute values in this case could be very low anyways (Fig6.).
In the MSC-THP-1 co-culture experimental setting, it would be much better to demonstrate the parallel increase of factors that consequently reduce inflammation (e.g. PGE2, IL-10) and provide absolute concentrations as well.
Furthermore, one should consider other type of experiments (inhibition of T cell proliferation). Functional in vitro tests should be relevant for later clinical use of the particular MSC product. The authors do not provide information on the future clinical use of the cells.

Validity of the findings

I have concerns about the high percentage of positivity with the CD146 marker (Fig4.a) that supposed to stain a subpopulation of MSCs as published by multiple groups earlier. The authors discuss this phenomenon as known in dental pulp MSCs cultured with serum and xenofree medium, but here they use MSCs from bone marrow and adipose tissue and they also have high CD146 positivity (30-95%) in their reference culture with alphaMEM medium and human concentrate platelet lysate.

Reviewer 2 ·

Basic reporting

No comment

Experimental design

No comment

Validity of the findings

No comment

Additional comments

The manuscript provides an interesting comparison of platelet lysate versus SXFM cultured MSCs. Since, SXFMs will be needed for the commercial production of MSCs, the present study is relevant for the advancement of the field. However, the methods, results, and discussion section require some major revisions before the manuscript can be accepted.

Introduction:
The introduction is adequate, but there are a few minor issues
• Line 90: What is the significance of specifically mentioning psoralen in the sentence?
• Line 93: The following sentence does not make good sense: ‘Finally, hCPL release criteria that have been recently proposed may be challenging for small structure and encouraging alternative products’

Methods:
• Figure 1 greatly helps in understanding the structure of the study. Although there is some ambiguity in the methods section that should be corrected.
• Multipotency assay: The text does not mention if hCPL or SXFM were added to the differentiation media. Moreover, it is mentioned that FBS was added to both the differentiation media. Please explain why?
• Immunomodulatory assay: not clearly mentioned at what step hCPL or SXFM was added in the experiment.
• Statistics: Ln 237, where were the non-inferiority analyses applied?

Results:
• Figure 2: Please provide images of the whole plate for CFU assay showing multiple colonies, in addition to the zoomed-in images of a single colony in 2A. That will help in visualizing the difference in number, size, and density of the colonies, as mentioned in the results.
• Figure 3: Statistical results should be presented; the level of negative markers seems variable between CPL and SXFM and could be significantly different. Please also explain how the authors decided that less than 5% positive cells for a maker is considered lack of expression (Ln 274). The figure legends states that 'N=2-6', is that a typographic error?
• Figure 4: Can the authors quantitatively compare the differentiation of cells in the 2 media (probably by quantify the stains?), if not please discus this limitation.

Discussion
Please discus the limitations of the present study and the future directions required to further validate the clinical utility of the SXFM under investigation.

Supplemental data
Explain the content of the supplemental data in the manuscript text or remove it. In its current form the supplemental does not make much sense.

Minor corrections:
• Ln 147: Replace ‘on the it’ with 'on them'
• Ln 172: centrifuged
• Ln 177: analyses
• Ln 217: should it be 0.2 and 1 x 10^6 cells/mL?
• Ln 343: immunomodulation
• Ln 344: 4 X 10^6 cells

---

## Round 0.2 · accepted · Accept

The authors have responded to the comments of the reviewers.

Reviewer 1 ·

Basic reporting

Manuscript 56142-2:
I accept the responses to my comments and modifications made by the authors.

Experimental design

N/A

Validity of the findings

N/A

Additional comments

No comments